# On the classification of Microsoft-Windows ransomware using hardware profile

Sana Aurangzeb[1,*], Rao Naveed Bin Rais[2,*], Muhammad Aleem[3], Muhammad Arshad Islam[3] and Muhammad Azhar Iqbal[4]

[1] Department of Computer Science, National University of Modern Languages, Islamabad, Islamabad, ICT, Pakistan
[2] College of Engineering and Information Technology, Ajman University, Ajman, United Arab Emirates
[3] Department of Computer Science, National University of Computer and Emerging Sciences, Islamabad, Islamabad, ICT, Pakistan
[4] School of Information Science and Technology (SIST), Southwest Jiaotong University, Chengdu, China
* These authors contributed equally to this work.



## ABSTRACT

Due to the expeditious inclination of online services usage, the incidents of ransomware proliferation being reported are on the rise. Ransomware is a more hazardous threat than other malware as the victim of ransomware cannot regain access to the hijacked device until some form of compensation is paid. In the literature, several dynamic analysis techniques have been employed for the detection of malware including ransomware; however, to the best of our knowledge, hardware execution profile for ransomware analysis has not been investigated for this purpose, as of today. In this study, we show that the true execution picture obtained via a hardware execution profile is beneficial to identify the obfuscated ransomware too. We evaluate the features obtained from hardware performance counters to classify malicious applications into ransomware and non-ransomware categories using several machine learning algorithms such as Random Forest, Decision Tree, Gradient Boosting, and Extreme Gradient Boosting. The employed data set comprises 80 ransomware and 80 non-ransomware applications, which are collected using the VirusShare platform. The results revealed that extracted hardware features play a substantial part in the identification and detection of ransomware with F-measure score of 0.97 achieved by Random Forest and Extreme Gradient Boosting.

## INTRODUCTION

Over the past several years, an exponential increase has been reported in ransomware attacks. Ransomware is the sub-class of malware that hijacks a device and blocks the victim to access the data until a compensation of some form is made. Typically, this compensation is in the form of money to concede access back to the victim. Ransomware has the ability to harmfully affect various kinds of devices such as *personal computers, servers, smartphones, tablets*, etc. For instance, multiple new variants of ransomware including WannaCry, JAFF, Petya have been reported in 2017 (*Hampton, Baig & Zeadall, 2018*).

Corresponding author
Rao Naveed Bin Rais,
r.rais@ajman.ac.ae

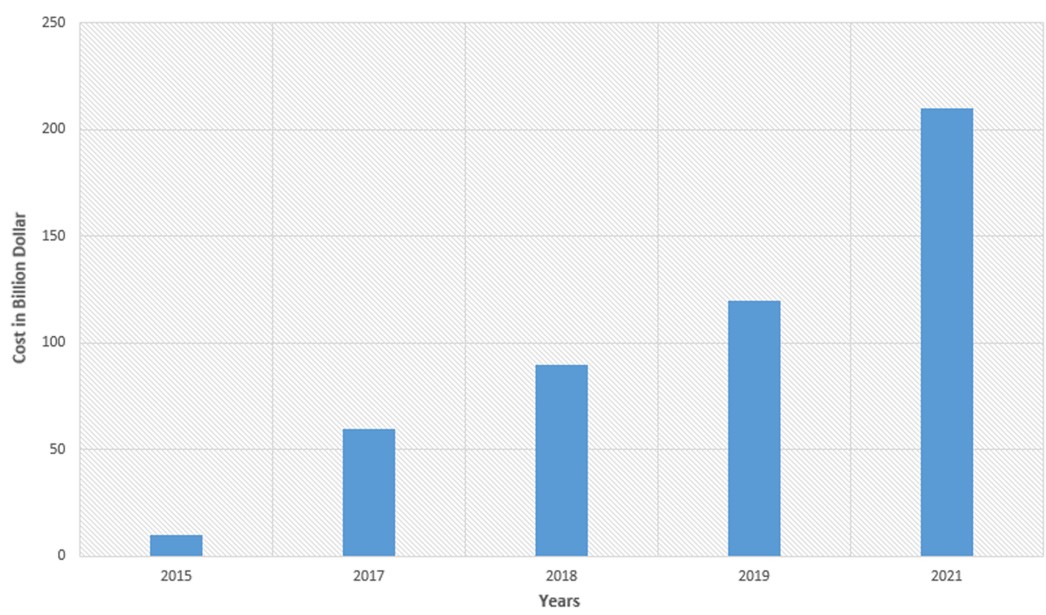

**Figure 1 Estimation and projection of losses (in Billions USD) caused by different ransomware between 2015 and 2021 (*Ramesh & Menen, 2020*; *Chung, 2019*).**

On 12 May 2017, within the span of a few hours, the WannaCry ransomware (*Maurya et al., 2018*) infected more than 70,000 desktop devices in over 150 countries across the globe (*Grant & Parkinson, 2018*).

The financial loss incurred due to ransomware can be quite devastating. For instance, CryptoWall_v3 ransomware (*Ali, 2017*; *Sgandurra et al., 2016*) caused the loss of an estimated $325 million in the US from November 2015 to June 2016.). In 2018, it was reported that 45% of the companies paid a ransom to recover the data stored on infected machines, which increased to 58% in the year 2019 (*Ramesh & Menen, 2020*). Another ransomware attack, triggered by CryptoWall_v4 ransomware resulted in a loss of $7.1 million worldwide (*Ali, 2017*). Recently reported ransomware attacks involving NotPetya and WannaCry are estimated to inflict the costs around $18 billion (*Davies, Macfarlane & Buchanan, 2020*). These attacks wreaked havoc in systems of various world organizations by halting and damaging their daily operations. It seems that losses caused by ransomware will probably exceed $20 billion by the end of the year 2021 (shown in Fig. 1) as reported by the Global ransomware-protection report (*Ramesh & Menen, 2020*; *Chung, 2019*).

Although, malware is deemed a great threat over the years, yet ransomware is an even more daunting threat compared to other malware due to its attacking and demanding nature (i.e., expecting a ransom in return). The classification of ransomware from traditional malware is essential because of their higher damaging impact in terms of informative data and financial loss. Compared to typical malware, it is challenging to identify and kill ransomware even when it is discovered, and the damage can be potentially irreparable even after its deletion *Al-rimy, Maarof & Shaid (2018)* and *Zhang et al. (2019)*. Hence, we require proactive and aggressive techniques to handle ransomware.

Moreover, it is very challenging to recognize and isolate the malware from ransomware due to the similarity in nature. A ransomware is more menacing than malware, as it not only damages the system and results in loss of control from the system but also demands compensation in return. Therefore, there is a need to have the proper distinction of ransomware from other malware (*Aurangzeb et al., 2017*; *Kok et al., 2019*; *Zhang et al., 2019*) to save billions of dollars in financial losses (*Davies, Macfarlane & Buchanan, 2020*).

Before analyzing the ransomware, one of the mandatory steps is the accurate identification of a particular type of ransomware and differentiating it from other typical malware. Broadly, malware analysis techniques are categorized as (1) *static* and (2) *dynamic* analysis (*Chen et al., 2017*). Besides, various researchers have employed the combinations of static and dynamic techniques in the form of hybrid analysis techniques. The procedure of scrutinizing a potential malware without executing the program is referred to as *static analysis*, whereas, the analysis performed via observing the execution behavior of a malware is known as *dynamic analysis*. Most contemporary state-of-the-art *dynamic analysis* techniques detect and classify ransomware that hides itself using various obfuscation techniques such as packed programs, compressed, or data transformation, indirect addressing, etc. (*Behera & Bhaskari, 2015*). A ransomware employs different hijacking strategies such as behaving like an adware resulting in unwanted advertisements or hiding itself using rootkits to bypass Anti-Viruses (AV) (*Demme et al., 2013*). A rootkit is a malware that alters the operating system (OS) and resides in the system for a prolonged period (*Aurangzeb et al., 2017*). Today, various anti-viruses tackle malware to dampen their caused and expected damages. However, the techniques employed by the anti-viruses are often limited to the prior knowledge (e.g., signatures, etc.) and there is a need to have more comprehensive *dynamic analysis* that could detect ransomware, employing the obfuscation techniques (*Demme et al., 2013*), utilizing hardware performance counters.

Hardware Performance Counters (HPCs) have been typically used by the programmers to analyze and measure the performance of applications and to identify the execution bottlenecks of a program (*Beneventi et al., 2017*; *Kuruvila, Kundu & Basu, 2020*). Initially, HPCs have been employed for investigating the *static* and *dynamic analysis* of programs to detect any malicious amendments as mentioned in *Alam et al. (2020)* and *Malone, Zahran & Karri (2011)*. Several studies (*Das et al., 2019*; *Demme et al., 2013*; *Singh et al., 2017*; *Wang et al., 2016*) discuss potential implications of using HPC for application analysis, and the majority of them suggest that hardware execution profile can effectuate the detection of malware (*Demme et al., 2013*; *Singh et al., 2017*; *Wang et al., 2016*; *Kuruvila, Kundu & Basu, 2020*). Another study (*Xu et al., 2017*) has utilized the hardware execution profiles to detect malware using machine learning algorithms, as malware changes data structures and control flow, leaving fingerprints on accesses to program memory. In this respect, they proposed a framework for detecting malware from benign applications that uses machine learning to classify malicious behavior of malware based on access patterns of virtual memory. *Zhou et al. (2018)* investigated whether HPCs

are useful in differentiating the malware from benign applications. However, the study did not consider malware as ransomware. However, utilizing the hardware performance measurements and the profile of the low-level execution behavior has not been previously studied for the analysis and detection of ransomware applications. We argue that ransomware reveals itself by exhibiting peculiar patterns in HPCs (e.g., through clock cycles, cache misses and hits, branch instructions and misses, retired instructions, etc.).

In this article, we present a framework based on dynamic analysis that mainly focuses on the classification of ransomware from non-ransomware. This article contemplates HPCs to detect Microsoft Windows-based ransomware by analyzing the execution behavior of ransomware. We primarily focus to determine the potential use of HPCs in analyzing and proactively detecting ransomware. Moreover, the classification of ransomware from malware is imperative because the damages caused by ransomware drastically ensure the data and monetary loss. To address this concern, we propose a mechanism that utilizes the application execution profile for the classification and detection of ransomware from non-ransomware. For classification, the application's hardware related performance features are extracted from the data set of 160 malware (consisting of 80 ransomware and 80 non-ransomware). Afterward, these features are fed to some well-known machine learning classification models such as Decision Tree (*Kohavi, 1996*), Random Forest (*Liaw & Wiener, 2002*), Gradient Boosting (*Friedman, 1999*) and Extreme Gradient Boosting (XGBoost) (*Chen & Tong, 2021*). These four classifiers are generally used for classification tasks of various applications including *spam detection*, *face recognition* and *financial predictions* (*Jordan & Mitchell, 2015*; *Kuruvila, Kundu & Basu, 2020*), etc. We employ these four classifiers as part of the proposed methodology to analyze their performance for ransomware detection. These models perform binary classification of malicious software into ransomware or non-ransomware classes. In summary, the main contributions of this article are as follows:

- In-depth analysis of the current state-of-the-art to identify the merits and demerits of several existing approaches;
- A novel mechanism for the classification and detection of malicious applications into ransomware and non-ransomware; and
- An empirical investigation of the HPCs against state-of-the-art dynamic techniques using machine learning classifiers;

The outcomes revealed that both the random forest and extreme gradient boosting classifier has outperformed decision tree and gradient boosting by attaining accuracy of 0.97 for classification.

The rest of the article is organized as follows. "Related Work" describes the related work. "Motivation and Methodology" presents the proposed methodology, dataset and feature extraction mechanism. In "Results and Discussion", the experimental setup details, results and related discussions are presented and "Conclusions" concludes the article.

## RELATED WORK

For *dynamic analysis*, it is necessary to collect key ransomware features at runtime. Most of the *dynamic analysis*-based research studies exploit the renowned malware databases (www.virusshare.com) for the acquisition of malicious software and use quarantine environments (such as Cuckoo's sandbox (*Kaur, Dhir & Singh, 2017*)) to execute the applications.

In *Zavarsky & Lindskog (2016)*, the authors presented an experimental analysis of Microsoft Windows and Android-based ransomware. This analysis demonstrates that ransomware detection could be performed by monitoring the abnormalities in the file system and registry activities. It is shown that a significant number of ransomware families exhibit very similar characteristics. Moreover, the authors concluded that changes in a particular set of registry keys are important aspects to be analyzed for ransomware detection. The authors discovered that Microsoft Windows 10 is reasonably effective against ransomware attacks. Moreover, this study also revealed that for the Android platform, the *Android Manifest* file and the permissions (required by an app) should also be considered for ransomware detection. There is a lot of work (*Alzahrani & Alghazzawi, 2019*; *Victoriano, 2019*) related to Android malware detection using machine learning approaches to classify malware families. Authors in *Scalas et al. (2019)* focus on ransomware classification and proposed a learning-based detection strategy. The proposed scheme relies on system's API information such as packages, classes, and methods related traces. The proposed scheme is capable to differentiate and classify generic malware, ransomware, and goodware. The experimental results highlight the significance and effectiveness of using system API information for Android ransomware classification.

Several researchers utilized the hash information (i.e., comparing hash values) to detect the CryptoLocker ransomware (*Song, Kim & Lee, 2016*). The affected systems are recovered by the following ways: (1) process CryptoLocker, (2) comparing hash information with the encrypted data files (3) validating the key using the key-index information stored therein and (4) proceeding to decode. Generally, this type of process consumes a lot of time for ransomware detection with a potential risk that another ransomware appears until a security company comes up with decryption keys of the old ransomware. Moreover, additional analysis is needed to detect new patterns of ransomware as the hackers persistently come up with new variants of ransomware. On the Android platform, another technique is proposed (*Song, Kim & Lee, 2016*) to prevent ransomware intrusion. The technique requires intense monitoring of the execution processes and analysis of the particular file directories using the statistical techniques, such as *Next-generation Intrusion Detection Expert System* (NIDES) (*Anderson, Thane & Alfonso, 1995*) using the *processor*, *memory usage* and *I/O rates*, to uncover the applications exhibiting abnormal behavior (*Song, Kim & Lee, 2016*).

Several other research studies have harnessed the machine learning-based approaches and dynamic or runtime features of executing applications to detect ransomware. Recently, HPCs-based events and their features are being used widely in research to detect side-channel attacks and ransomware (*Or-Meir et al., 2019*). *Alam et al. (2020)* have used

HPCs features to detect malware from benign applications. The authors proposed an anomaly detection technique to identify malicious ransomware in a few seconds with very few false positives using Recurrent Neural networks (RNN). However, only five hardware performance measures that is, instruction, cache-references, cache-misses, branches and branch-misses are investigated, whereas the authors investigation is with one type of ransomware only, which was WannaCry. In *Kadiyala et al. (2020)*, only four hardware performance aspects were considered. *Maiorca et al. (2017)* proposed a supervised machine learning-based procedure, *R-PackDroid*, to detect Android ransomware, which is a light-weight technique and does not require prior knowledge of ransomware's encryption mechanisms. However, the R-PackDroid technique uses fully encrypted code-files and is unable to analyze the applications that load the code at run-time. The R-PackDroid can be incorporated with the other *dynamic analysis* methods, such as the approach proposed by *Kimberly et al. (2015)*. Moreover, R-PackDroid based application analysis strongly depends on the parsing capabilities of the *ApkTool* framework.

*Narudin et al. (2016)* has presented a machine learning-based malware analysis approach based on the anomaly detection mechanism. The results indicated that Bayes network and Random Forest classifiers produce accurate results by attaining 99.97% *True-Positive Rate* (TPR) as compared to the multi-layer perceptron technique with only 93.03% TPR using the *MalGenome* data set. However, the accuracy of this scheme dropped to 85% for the latest malware experiments.

Desktop ransomware can easily bypass any counter-measures and thus results in the seizure of personal data. Authors (*Al-rimy, Maarof & Shaid, 2018*) presented an effective mechanism for early diagnosis and avoidance of the crypto-ransomware, which is based on machine learning techniques (*One-Class SVM* and n-gram technique (*Zhang, Xu & Wang, 2015*)) and comprises three modules: (1) pre-processing, (2) features engineering and (3) detection module. The authors employed an adaptive anomaly detection mechanism that handles the dynamic characteristics of systems and frequently updates the normal profile built from the feature extraction (*Al-rimy, Maarof & Shaid, 2017*) to improve the accuracy of detection.

The study (*Kharraz et al., 2015*) has presented the analysis of ransomware families (the year 2006–2014) and concludes that the suspicious activity of file systems should be observed for ransomware detection. For instance, the changes in the types of *I/O Request Packets* (IRP) or the *Master File Table* (MFT) are usually formed to access the file system. A considerable number of ransomware families share related features as a core part of the attacks; however, there still lacks a reliable destructive function to successfully infect files of victims. In Table 1, we recapitulate several other prominent ransomware detections (*Yang et al., 2015*; *Andronio, Zanero & Maggi, 2015*; *Kharraz et al., 2016*) and prevention (*Ahmadian, Shahriari & Ghaffarian, 2015*; *Kim, Soh & Kim, 2015*; *Lee, Moon & Park, 2016*; *Brewer, 2016*) techniques. Recently, deep neural networks and convolutional neural networks (CNNs) have shown remarkable performance in the area of object recognition (*Simonyan & Zisserman, 2014*). The deep convolutional neural networks can outperform the other approaches like Natural Language Processing (NLP),

**Table 1 Comprehensive Comparison of the state-of-the-art approaches along with their key points, drawbacks and implementation design approach.**

| References | Methodology | Strengths | Limitations |
|---|---|---|---|
| Demme et al. (2013) | • Dynamic approach<br>• Android Malware detection with performance counters<br>• Applied ML algorithms (KNN, Decision tree) | • Major support is that runtime behavior can capture using HW performance counters are essential to detect malware90% accuracy with 3% FP | • Able to detect some variants whereas some were not detected<br>• Malware label data might not accurate |
| Kharraz et al. (2015) | • Analyzed 15 ransomware families<br>• Proposed various mitigation approaches to decoy resources to detect malicious file access. | • Provide evolution-based study of RW attacks from a long-term study 2006-2014<br>• Detailed analysis of Bitcoin for monetization | • Assumed that every file system access to delete or encrypt decoy resources<br>• However, they didn't implement any concrete solution to detect or defend against these attacks |
| Kim, Soh & Kim (2015) | • Present a quantification model based on social engineering technique to avoid and identify any cryptographic operations in the local drive | • explains the file-based intrusion detection system and IP traceback algorithm | • Lack of experimental results<br>• Suggests guidelines online |
| Narudin et al. (2016) | • Machine learning-based study<br>• Filter TCP packets, extract network traffic features<br>• Evaluate Bayes, Random Forest, KNN, J48, & MLP | • Accurate detection based on ML classifiers.<br>• BN and RF produces 99.97% TPR<br>• Bayes, MLP with ROC 0.995 and RF with 0.991 | • Applicable for Android platform only |
| Zavarsky & Lindskog (2016) | • the life cycle of Windows-based Ransomware study.<br>• Implement basic static and basic dynamic<br>• MD5 method, Cuckoo Sandbox used.<br>• For android Analyze AndroidManifest.xml, administrative privilege<br>• For Windows analyze Filesystems, registry activities, and network operations | • Explained the detailed analysis, working, and functionality of Ransomware<br>• Performed analysis on both the Windows and Android-based RW<br>• PEiD tool is used for windows ransomware detection | • Performed only basic static and dynamic analysis.<br>• No machine learning-based approach to detect zero-day ransomware<br>• Lack of experimental analysis |
| Song, Kim & Lee (2016) | • Proposed techniques on three modules: Configuration, Monitors, and Processessing<br>• the hash information method is used for detection of CryptoLocker type ransomware | • The proposed technique monitors the processes and specific file directories<br>• monitor file events using statistical methods on Processor usage, Memory usage, and I/O rates | • Not applicable for Windows-based ransomware<br>• No classifier is used<br>• Does not install applications and execute for prevention and detection<br>• Results are not analyzed quantitatively |
| Kharraz et al. (2016) | • dynamic approach<br>• Monitors file system I/O activity<br>• Detect screen locking mechanism,<br>• used Tesseract-OCR | • new ransomware family were detected that was not detected previously<br>• The long-term study analyzed 148223 malware samples and correctly detect and verified 13637 ransomware samples<br>• 96.3% TP rate and 0 FPs | • Accuracy is not that good. For example, the system correctly detects 7,572 ransomware whereas only one unknown was detected |

(Continued)

| References | Methodology | Strengths | Limitations |
|---|---|---|---|
| *Sgandurra et al. (2016)* | • Dynamically monitor file system activity on windows platform<br>• Classify between goodware and ransomware using ML<br>• Mutual Information and Regularized Logistic Regression classifier used.<br>• Proposed machine learning approach EldeRan | • effective and entirely automated tool to analyze new software and enhance the detection capabilities of AV software<br>• registry key and API calls are the two classes with the most relevant features.<br>• EldeRan achieves ROC curve of 0.995, detection rate 96.3% | • Despite good results, EldeRan still not be used as a replacement for AV<br>• the current settings have no other applications running in the VM, except the ones coming with a fresh installation of Windows,<br>• initial dataset was larger<br>• Unable to analyze RW that shows silent behavior, or wait for the user to do something |
| *Chen & Robert (2017)* | • Dynamic behavioral analysis of wanna cry<br>• Present a method to extract features of malware from hosts logs<br>• TF-IDF approach gives better results for analyzing wanna cry | • Research helps in further manual analysis of logs from ambient system logs in forensic efforts.<br>• Automatically generate behavior analysis of malware samples from sandbox log data | • Presentation and experimented results are outside the scope of the article<br>• Study not help in analyzing automatic pattern generation |
| *Al-rimy, Maarof & Shaid (2017)* | • Machine learning n-gram, EFCM,<br>• Information Gain,<br>• Sliding window<br>• Static + dynamic conf<br>• SVM for behavioral detection | • Proposed framework inclines to share the pre-encryption data space as the main defense step against crypto-ransomware attacks | • No classification<br>• No experimental work<br>• No results evaluation details |
| *Bahador, Abadi & Tajoddin (2019)* | • Presents a two-stage heuristic matching strategy signature-based approach to hardware-level behavioral malware detection and classification | • HLMD approach can detect malicious applications at the beginning of the execution and can achieve an average precision, recall, and F-measure of 95.19%, 89.96%, and 92.50%, respectively | • Their approach is suitable for independent malicious programs (worms, Trojans and bots) that can be run standalone without having to be attached to a host program<br>• Not applicable for Ransomware |
| *Dion & Brohi (2020)* | • analyzed the opcodes and measures their frequencies.<br>• Compare the performance of supervised machine learning algorithms for ransomware classification | • Experimental analysis of Random Forest, Gradient Boosting Decision Tree (GBDT), Neural Network using Multilayer Perceptron and three types of Support Vector Machine (SVM) were performed<br>• Random Forest and GBDT outperformed | • Authors mentioned that the experimental platform can be able to identify only exe or ddl format ransomware<br>• Only supervised machine learning applied |
| *Kadiyala et al. (2020)* | • Malware Analysis using Hardware Performance Counters<br>• Proposed a three-step methodology included i) extracting the HPCs ii) finding maximum variance through reducing fine-grained data iii) apply ML algorithms | • extract the HPCs for each system call during the runtime of the program using perf libraries along with CoreSight Access Libraries that allows to interact directly through APIs<br>• detection rate 98.4% | • suitable for linux environment<br>• Training set is small<br>• Monitored only four hardware performance counters<br>• 3.1% false positive |

| References | Methodology | Strengths | Limitations |
|---|---|---|---|
| *Alam et al. (2020)* | • Dynamic Analysis<br>• Implement Artificial Neural Network and Fast Fourier Transformation<br>• Disk encryption detection module process used | • Two-step detection framework named as RAPPER<br>• an accurate, fast, and reliable solution to detect ransomware.<br>• Used minimal tracepoints<br>• Provide a comprehensive solution to tackle standard benchmark,<br>• disk encryption and regular high<br>• computational processes<br>• HPCs were used to analyze files using perf tool | • Observe 5 events of HPCs only i.e., instruction, cache-references, cache-misses, branches, and branch-misses<br>• Analyze and present all the case studies by giving a comparison with WannaCry only<br>• Lack of detailed experimental results and accuracies. |
| Our Approach | • Dynamic Analysis of Hardware Performance counters<br>• Performed classification techniques on Windows-based executable files | • Apply ML algorithm such as RF, Decision Tree, Gradient Boosting, Extreme Gradient Boosting<br>• Attained F-measure score of 0.97<br>• Random Forest and Extreme Gradient boosting outperformed | • Dataset was initially large but after preprocessing remain small dataset<br>• Only supervised machine learning techniques applied |

if the training is performed using large datasets (*Liu & Liu, 2014*; *Zhang, Zhao & LeCun, 2015*). Due to the limited dataset, we have employed supervised machine learning algorithms. Moreover, the core objective of our proposed scheme is to gauge the effectiveness of the hardware execution profile (i.e., a truly dynamic environment) for the classification of ransomware/non-ransomware.

Besides, the performance counters exhibit the true application execution behavior and are being employed by the researchers to analyze application performance (*Mucci et al., 1999*; *Bahador, Abadi & Tajoddin, 2014*; *Demme et al., 2013*). In *Basu et al. (2019)* authors have used hardware performance counters to detect Android malware, and in another similar work (*Bahador, Abadi & Tajoddin, 2019*) authors have presented a heuristic (using signature-based features and hardware performance counters) to detect and classify malware. Their approach is only suitable for malware detection that are invoked as standalone applications and are not dependent on other host applications. In summary, none of the existing *dynamic analysis* techniques utilizes the important dynamic feature such as HPCs to detect Windows platform based malicious applications. Although, there are few approaches available that classify a benign application from ransomware, however, to the best of our knowledge no other approach (utilizing hardware performance counters) classified malware into the subclass of ransomware/non-ransomware on the Windows platform. Malware can employ obfuscation techniques to deceive static analysis based anti-viruses. Furthermore, runtime behavior cannot be obfuscated and can be detected using dynamic analysis. We believe this aspect should essentially be exploited and the hardware execution profile should be utilized to execute applications for ransomware detection. Based on these facts, we argue that HPCs are useful features that could be utilized for the detection and classification of ransomware. In this study,

we employ various machine learning classifiers such as Decision Tree, Random Forest, Gradient Boosting and Extreme Gradient Boosting along with the HPCs to address the following questions:

1. How different are ransomware from malware at runtime considering machine learning techniques?
2. Which of the hardware performance counters (HPCs) play a vital role in ransomware detection?

## MOTIVATION AND METHODOLOGY

The dynamic analysis holds adequate potential to accurately detect the threat of ransomware because an executable program cannot hide its true characteristic. Therefore, most of the anti-virus vendors rely on automated dynamic analysis mechanisms to detect new variants of ransomware. Most of the antiviruses apply the heuristics combined with the behavior analysis to deduce whether an executable is *benign* or *malware* (*Sgandurra et al., 2016*).

A wide range of HPCs that is, *clock cycles, cache hits, cache misses, branch instructions, branch misses, retired instructions*, etc. are used to observe the behavior of an executing application (*Chiappetta, Savas & Yilmaz, 2016*). Usually, the symmetric encryption marks the cache-based events while the asymmetric encryptions do have an impact on the instruction and branching events as explained in *Alam et al. (2020)*. HPCs have been harnessed by many application developers to identify the computation and memory bottlenecks to improve the performance and reliability of the executing applications (*Chiappetta, Savas & Yilmaz, 2016*). In this study, we utilize 11 performance counters for the classification of ransomware. For classification, we train the employed machine learning classifiers to analyze the dynamic behavior of ransomware and non-ransomware malicious programs. Moreover, the classification of Ransomware from *Traditional Malware* is essential due to the intensity of the damage caused in terms of financial loss. Unlike traditional malware, it is more troublesome to identify and kill ransomware even when it is discovered, and the damage is irreparable even after its removal *Al-rimy, Maarof & Shaid (2018)* and *Zhang et al. (2019)*. Hence, it is very important to recognize and isolate the malware from ransomware due to the similarity in nature. Therefore, it is required to devise a formal classification mechanism to discriminate ransomware from other non-ransomware *Zhang et al. (2019) Aurangzeb et al. (2017)* and *Kok et al. (2019)* to avoid billions of transactions in the name of ransom.

### Dataset collection

For the experimentation, we have obtained randomly selected 160 Windows-based malware from *VirusShare*. VirusShare repository provides the dataset related to ransomware and many other types of malicious applications of the Windows platform (in addition to the other platforms such as Android, Linux, etc.). It is frequently updated and at presently contains the latest malicious applications contributed by the community (*Kouliaridis & Kambourakis 2020*). Due to the diversity, the *VirusShare* platform is

very popular in the research community. We collected the dataset from *VirusShare* related to Windows-based malicious applications. After static analysis of the downloaded applications, obfuscated applications are eliminated. Afterward, each malware is labeled as a non-ransomware or ransomware based on the analysis data provided by many renowned anti-viruses available via *VirusShare*. These labels are further validated with the tags available from *VirusShare* for the sake of confirmation. In this study, benign binary files are not considered because the main aim of the study is to classify between ransomware and other malicious applications. Therefore, we consider the malicious applications category *Trojan* (as a non-ransomware sample) due to their similarity in activities with the ransomware (*Gazet, 2010*). The employed classifiers are trained using the behavioral features for ransomware and non-ransomware with explicit labeling (i.e., Ransomware/Non-Ransomware). Furthermore, a disjoint data set is used for training and testing purposes.

## Feature extraction

All malware in the data set are executed in a quarantine environment and their data related to hardware performance counters are collected using *perf* (an instrumentation and performance analysis tool (*De Melo, 2010*; *Weaver, 2013*; *Alam et al., 2020*)). To ensure the reliability and accuracy of the results, the mean values of three rounds of experiments are reported. We executed each application three times in a virtual machine (i.e., VMWorkstation 12 Pro 12.1.1 build 3770994) for no longer than 240 s with different input parameters to emulate a real interactive environment. After the execution of each malicious application, the virtual machine is reset to its original state using the snapshot feature (to ensure the performance counter trace collected during the previous execution do not intermingle with the current execution).

For binary classification problem discussed above, we employ hardware performance counters as features, that is, (1) *task clock*, (2) *context switching*, (3) *CPU utilized*, (4) *CPU migrations*, (5) *page faults*, (6) *CPU cycles*, (7) *cache-misses*, (8) *instructions retired*, (9) *branches taken*, (10) *branch-misses* and (11) *execution time*, (illustrated in Table 2) to train the machine learning classifier. We have executed ransomware applications on a PC within a virtual machine and recorded the features (i.e., hardware performance counters, etc.) using *perf*. The perf library provides the hardware performance counters related values representing the involvement of the several important hardware features of the processor during execution. Feature selection plays a significant role in achieving precise training of the employed machine learning models; thereby attaining accurate results with efficient performance and low overhead (*Li et al., 2017*). A correlation matrix among the employed features is generated to analyze the pattern that leads to the selection of features. Two features are considered negatively correlated if a change of one feature inversely impacts the value of the other feature. The features correlation analysis is presented in Fig. 2. If two numerical features are highly correlated, then one of them can be ignored. Therefore, we employed a sub-set of those features which are not co-related to reduce the computation overhead during the training process of the machine learning models. Figure 2 shows that the *Cache Misses* related hardware feature have a low positive

| Table 2 | Features set used in this work for performance evaluation (HPCs). | |
|---|---|---|
| S.no | Hardware features | Description |
| 1 | Task-clock | The task-clock shows the amount of time spent on the task (*Kuznetsova et al., 2017*) |
| 2 | CPU utilization | CPU-clock is based on the total time spent on the CPU |
| 3 | Context switching | explains how many times the software switched off the CPU from one process/thread to another (*Kuznetsova et al., 2017*) |
| 4 | CPU migration | CPU migration describes equality in a workload distribution across all cores. (*Kuznetsova et al., 2017*) |
| 5 | Page faults | Page-faults occur when a program's virtual content has to be copied to the physical memory (*Kuznetsova et al., 2017*) |
| 6 | Instructions per cycle | The average number of instructions executed for each clock cycle |
| 7 | Branch | A branch is an instruction in a computer program that can cause a computer to begin executing a different instruction sequence and thus deviate from its default behavior of executing instructions in order |
| 8 | Branch misses | Branch misprediction occurs when a processor mispredicts the next instruction to process in branch prediction, which is aimed at speeding up execution. |
| 9 | Cycles | Perf-CPU-cycles is a count of CPU cycles that traces to a hardware counter (*Flater, 2014*) |
| 10 | Cache misses | Cache misses is a state of not getting data that is being processed by a component or application that is not found in the cache. |
| 11 | Total time elapsed | It is the total execution time in seconds |

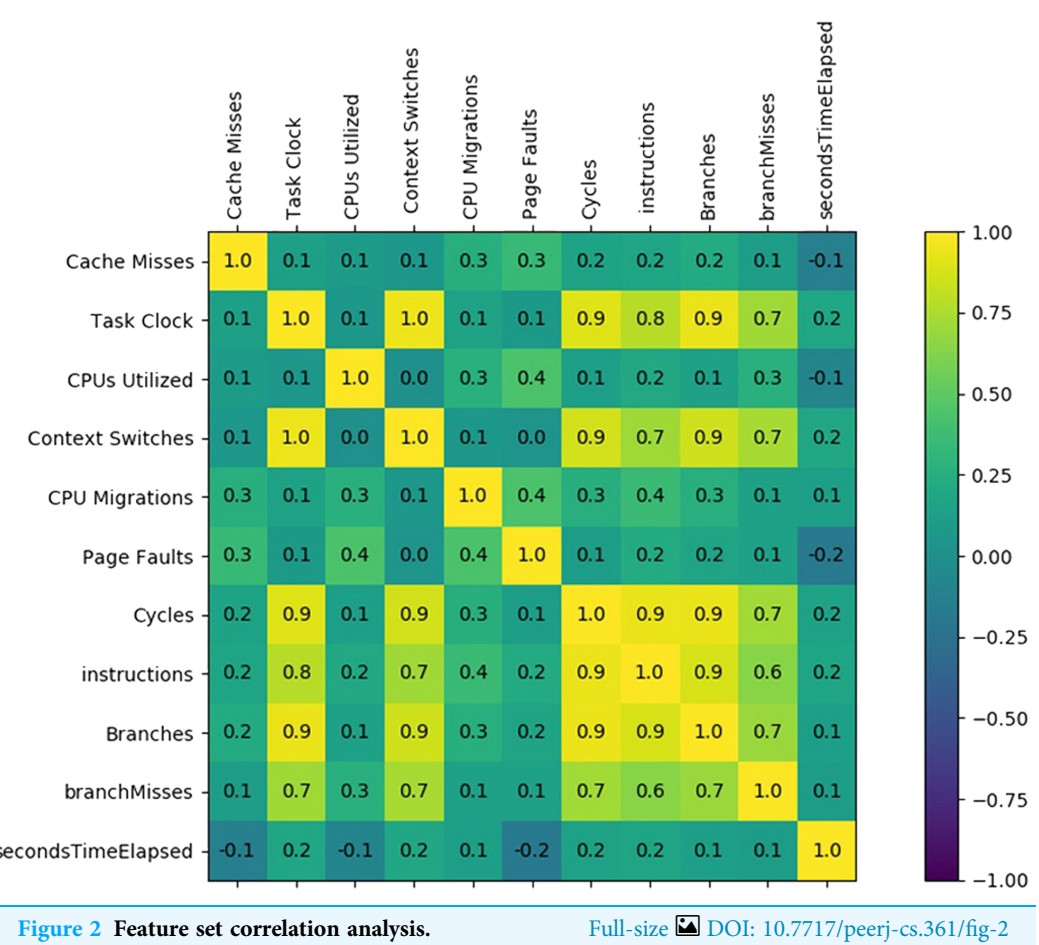

**Figure 2** Feature set correlation analysis.     

**Table 3 Features rank list.**

| Rank | Score | Feature |
|------|-------|---------|
| 1 | 0.20145 | Cache misses |
| 2 | 0.181887 | TaskClock |
| 3 | 0.153562 | Branches |
| 4 | 0.10867 | SecondsTimeElapsed |
| 5 | 0.086973 | Instructions |
| 6 | 0.085666 | BranchMisses |
| 7 | 0.044272 | ContextSwitches |
| 8 | 0.042727 | PageFaults |
| 9 | 0.040087 | CPU migration |
| 10 | 0.028564 | Cycles |
| 11 | 0.026142 | CPUsUtilized |

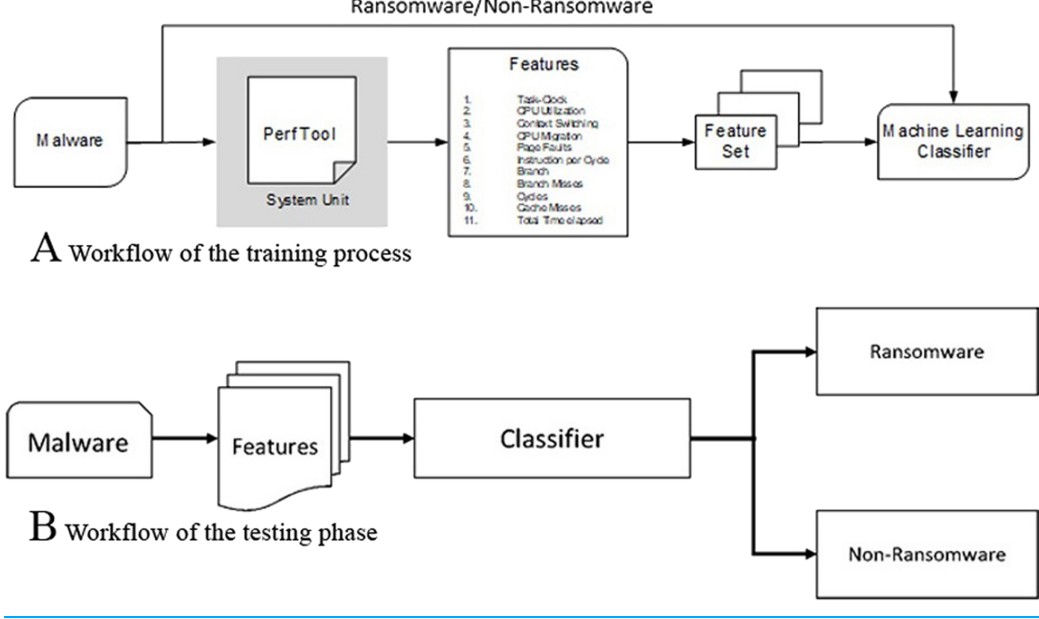

**Figure 3 Feature extraction workflow for training and testing phases: (A) Workflow of the training process, (B) workflow of the testing phase.**

correlation with all the other features. On the other hand, the Task Clock feature has a strong relationship with the *Context Switches, Cycles, Instructions, Branches*, and *Branches Misses*. The features having higher rank are deemed as potential features for classification than low ranked features as shown in Table 3.

In the training phase, hardware features are extracted by executing known malware and non-malware application in containing environment system units as shown in Figs. 3A and 3B. A total of 80% of the employed data set is used for training and 20% is used for testing. We have used *K*-fold (*k* = 10) cross-validation mechanism and compare the ransomware detection accuracy of different classifiers to make sure that the dataset is used uniformly without any biasness. This results in un-biased training and testing cycles

producing the results on which we could conclude with confidence. For each cycle of the training/testing, a 20% testing and 80% training partition was employed. The goal of supervised machine learning techniques is to find a function that is trained using the employed features such that the error is minimum for the new or unseen data. In the training phase, the classification model is trained using the HPCs as shown in Table 2. The testing or validation methodology is performed after the training of the classifiers.

## Classification model

The machine learning classification algorithms namely *Decision Tree, Random Forest, Gradient Boosting* and *Extreme Gradient Boosting* are used for classification purposes such as *phishing detection, facial recognition* and *financial predictions* (*Jordan & Mitchell, 2015*), etc. We employ these four classifiers as part of the proposed methodology to analyze their performance for ransomware detection.

The *decision tree* is a tree-based classifier, which contains a root, internal nodes, and leaf nodes. The class label is assigned to each leaf node and the decisions are rendered by the internal nodes (*Tan, Steinbach & Kumar, 2006*). *Random Forest* (RF) classifier is based on a combination of multiple decision tree predictors such that each tree depends on the values of a random vector sampled independently and with the same distribution for all trees in the forest (*Xuan et al., 2018*). The *Extreme Gradient Boosting XGBoost* and *Gradient Boosting* follow the same basic principle however, there are a few differences in their modeling details. Specifically, extreme gradient boosting utilizes a more regularized model formalization to control the *over-fitting problem* that may occur due to linear fitting over noisy data to provide better performance (*Jbabdi et al., 2012*). For the Decision Tree and Random Forest, the maximum tree depth is set as 2 to ensure that under-fitting issues are avoided. To achieve a smoother curve, the bagging technique is applied to the Random forest mechanism where each of the trees executes in a parallel way thus making a forest. As each tree is independent, therefore, the whole forest result is taken for the analysis (resulting in a smoother curve) To avoidover-fitting issues, we have evaluated our proposed technique using $K$-fold ($k = 10$) cross-validation. The first fold is evaluated with the other folds and the second time it executes, it takes the first and second fold to be compared with the rest, this goes on until 80% of training data is compared against 20% of the test data.

## RESULTS AND DISCUSSION

For experimentation, we utilize a system with Intel core i7 processor, 8 GBs of memory, and Ubuntu 12.10 OEM as an operating system. For classification, a machine learning tool *Scikit-learn* (*Pedregosa et al., 2011*; *Black et al., 2020*), is employed. To evaluate the results, standard evaluation measures that is, *precision, recall*, and *F-Measure* are calculated to determine the accuracy of each classifier. Equations (1)–(4) provide the mathematical description of accuracy, precision, recall, and f-measure, respectively. The terms used in Eqs. (1)–(4) are explained as follows: *True Positive* (TP) rate shows the number of predicted positives that are correct, while the *False Positive* (FP) rate refers to the number of predicted positives that are incorrect. Similarly, *True Negative* (TN) rate

shows the number of predicted negatives that are correct while the *False Negative* (FN) rate refers to the number of predicted negatives that are incorrect. The recall is the sensitivity for the most relevant result. F-measure is the value that estimates the entire system performance by calculating the harmonic mean of precision and recall. The maximum value of 1.000 for accuracy precision and recall indicates the best result (*Narudin et al., 2016*).

$$\text{Accuracy} = \frac{TP + TN}{TP + TN + FP + FN} \tag{1}$$

Precision denotes the proportion of Predicted Positive cases that are correctly Real Positives.

$$\text{Precision} = \frac{TP}{TP + FP} \tag{2}$$

The recall is the proportion of Real Positive cases that are Predicted Positive

$$\text{Recall} = \frac{TP}{TP + FN} \tag{3}$$

$$\text{F-Measure} = 2 * \frac{(\text{Precision} * \text{Recall})}{(\text{Precision} + \text{Recall})} \tag{4}$$

*Receiver Operating Characteristic* (ROC) curves (*Metz, 1978*; *Dion & Brohi, 2020*) are extensively being applied in significant researches to measure the accuracy of the machine learning models that are being trained to achieve actual performance (*Bradley, 1997*). Furthermore, ROC curves are applied in numerous systematic approaches that merge multiple clues, test results, etc., and are plotted and evaluated to characterize a qualitative feature of the particular. ROC is a plot wherein *Y*-axis is reserved for *True Positive Rate* (TPR) and *X*-axis is reserved for *False Positive Rate* (FPR). For all possible classifications such as the output class, the TPR rate depends on the set-up where the real classification is considered to be as positive and the number of times the classifier has predicted the result to be as positive. The FPR can be defined as how the classifier incorrectly labeled positive to those that are classified to be negative. Together the TPR and FPR values lie in-between 0 and 1 (1 indicating an accurate prediction).

The results based on the decision tree classifier can be seen in Fig. 4. The ROC curve for both classes (i.e., ransomware as class "1" and non-ransomware as class "0") is the same having value of 0.94 which signifies the excellent prediction. However, the precision-recall curve for class 0 that is, for *Non-Ransomware* shows accuracy of 0.89 or 89% whereas for class 1 that is, ransomware the accuracy is 0.93. The F-measure score of the Decision Tree is 0.94 as shown in Table 4.

The results obtained using the Random Forest classifier for two classes (i.e., *ransomware* and *non-ransomware*) are shown in Fig. 5 and F-measure score is illustrated in Table 5. The higher accuracy results are evident from the similar ROC curve value that is, 0.99 for both the ransomware and non-ransomware classes.

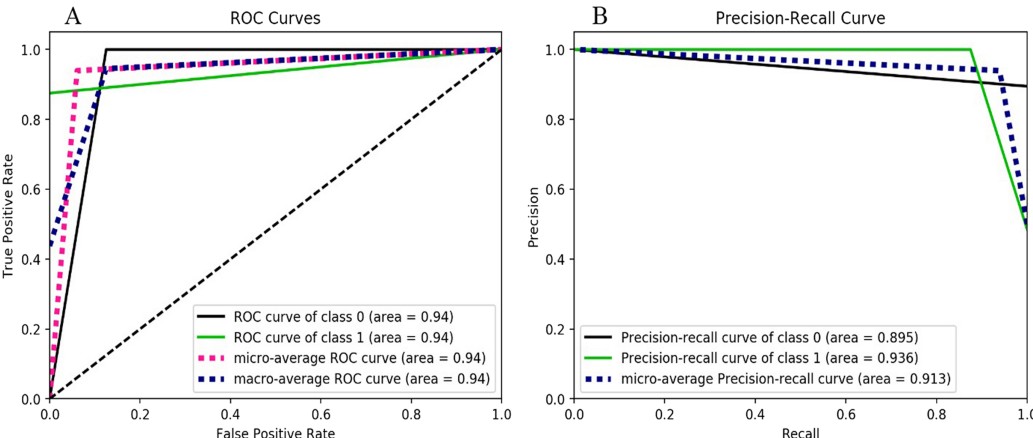

**Figure 4** Decision-tree performance metrics behavior, (A) ROC curves for the classes 0 and 1, (B) Precision-Recall curve for the classes 0 and 1.

**Table 4 Decision tree precision, recall and F-measure score for malware classes (0, 1).**

| Malware class | Precision | Recall | F-measure |
| --- | --- | --- | --- |
| Ransomware (class label 1) | 1.0 | 0.88 | 0.93 |
| Non-Ransomware (class label 0) | 0.89 | 1.0 | 0.94 |

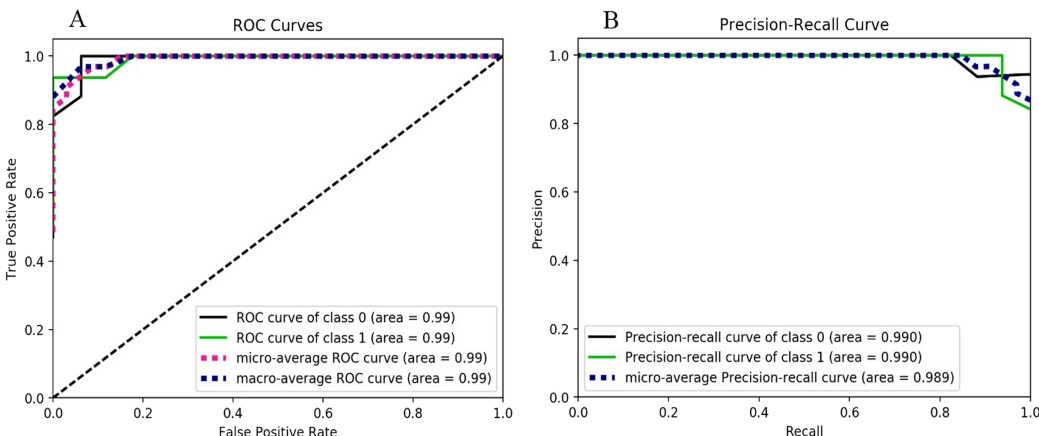

**Figure 5** Random forest performance metrics behavior, (A) ROC curves for the classes 0 and 1, (B) Precision-Recall curves for the classes 0 and 1.

**Table 5 Random forest precision recall and F-measure score against classes 0 and 1.**

| Malware class | Precision | Recall | F-measure |
| --- | --- | --- | --- |
| Ransomware (class label 1) | 1.0 | 0.94 | 0.97 |
| Non-Ransomware (class label 0) | 0.94 | 1.0 | 0.97 |

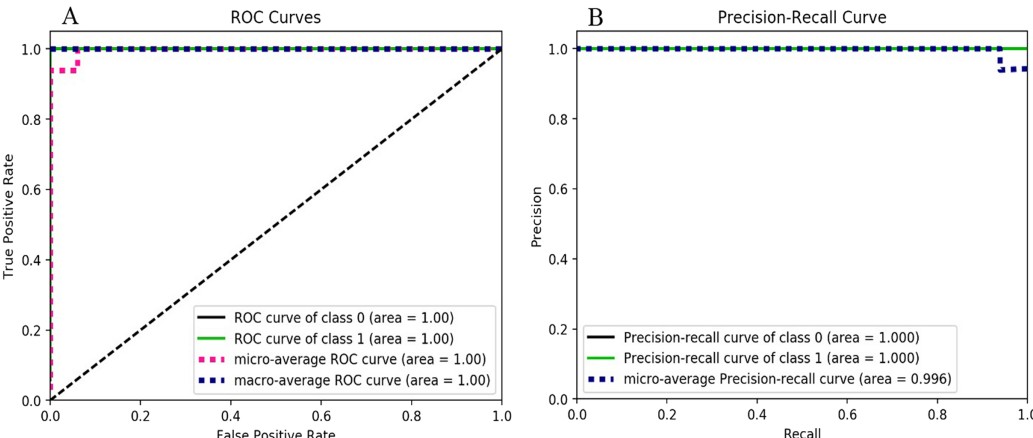

**Figure 6  Gradient boosting performance metrics behavior: (A) ROC curves for the classes 0 and 1. (B) Precision-Recall curves for the classes 0 and 1.**

**Table 6  Gradient boosting precision, recall and F-measure score for malware classes.**

| Malware class | Precision | Recall | F-measure |
|---|---|---|---|
| Ransomware (class label 1) | 1.0 | 0.88 | 0.93 |
| Non-Ransomware (class label 0) | 0.89 | 1.0 | 0.94 |

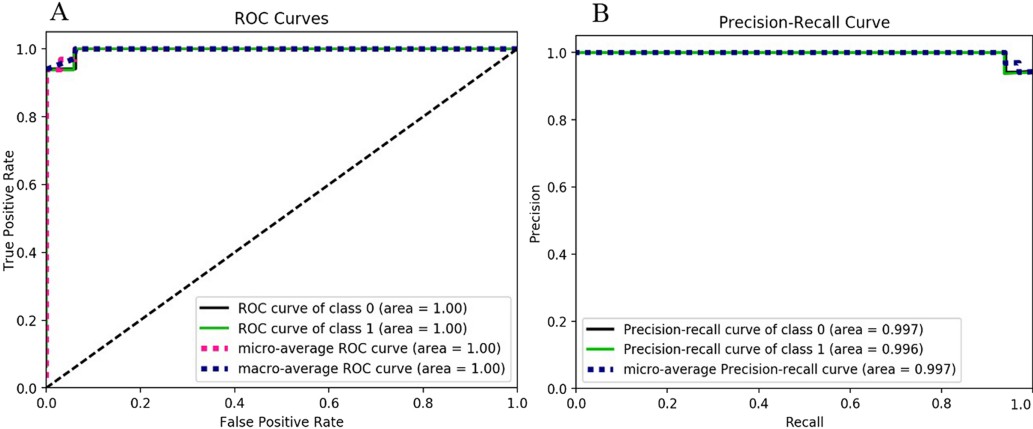

**Figure 7  Extreme gradient boosting performance metrics behavior: (A) ROC curves for the classes 0 and 1. (B) Precision-Recall curves for the classes 0 and 1.**

The results depicting the performance of gradient boosting, shown in Fig. 6, reveal that the ROC curve values for both the classes (i.e., *ransomware* and *non-ransomware*) as well as (i.e., 1.0) the precision-recall curves of both classes follow similar pattern of high accuracy. The F-measure score of the gradient boosting classifier is 0.93 for ransomware and 0.94 for non-ransomware (as shown in Table 6).

The extreme gradient boosting classification model-based results are shown in Fig. 7 and Table 7. The ROC curve and Precision-Recall Curve of both classes (i.e., ransomware and non-ransomware) are the same (i.e., 1.0). The extreme gradient boosting based model's F-measure score is 0.97, which is similar to the gradient boosting and random

**Table 7 Extreme gradient boosting precision, recall and F-measure score for malware.**

| Malware class | Precision | Recall | F-measure |
|---|---|---|---|
| Ransomware (class label 1) | 1.0 | 0.94 | 0.97 |
| Non-Ransomware (class label 0) | 0.94 | 1.0 | 0.97 |

**Table 8 Four classifiers result and their comparison F-measure score.**

| Classifier | F-measure |
|---|---|
| Decision Tree | 0.94 |
| Random Forest | 0.97 |
| Gradient Boosting | 0.94 |
| Extreme Gradient Boosting | 0.97 |

forest-based classification as shown in Table 8. The Random Forest-based classification model outperformed decision tree-based classification by attaining the accuracy of 0.94, as shown in Table 8. However, the value of the F-measure for both the classes is 0.97 (as shown in Table 8). The model has attained an improvement of 3% than the decision tree-based classification. The model shows similar F-measure results of 0.97 as observed for random forest and extreme gradient boosting.

This study has demonstrated the possibility of exploiting HPCs as the potential features for ransomware detection. After analyzing the sets of ransomware and non-ransomware, the features obtained from HPCs have been analyzed to classify malicious applications into ransomware and non-ransomware categories using several machine learning algorithms such as *Decision Tree, Random Forest, Gradient Boosting* and *Extreme Gradient Boosting*. The results of detailed experiments as stated earlier in the section have revealed that extracted hardware features play a significant role in the detection and identification of ransomware. Among all the employed machine learning classifiers, the random forest-based model and extreme gradient boosting have outperformed by yielding F-measure score of 0.97 followed by a decision tree that achieved 0.94 F-measure. Moreover, the features cache misses, task clock, and branches obtained through HPCs could be deemed as potential parameters in classifying ransomware from non-ransomware.

## CONCLUSIONS

In this article, the analysis of HPCs has been presented for Windows ransomware classification. The results have revealed that the HPCs hold the considerable potential to expose hidden indicators of the executing applications such as malicious codes and ransomware. Performance counters, that is, *cache misses, task clock* and *branches* have played a pivotal role in classifying ransomware in a way that if there are a high number of cache misses or a high number of branch mispredictions (where control flow becomes detectably anomalous) are good indicators that help in indicating a potential attack (*Foreman, 2018*). The proposed technique holds adequate potential to provide sufficient

detection accuracy by attaining the F-measure score of 0.97. This study demonstrated the possibility of exploiting HPCs as the potential feature for the detection of ransomware. However, this topic needs further investigation. In the future, we intend to scrutinize other dynamic features with the combination of call graphs to detect and classify ransomware. Moreover, the application of machine learning algorithms has shown very promising results in ransomware detection. In the future, we will expand this study to perform in-depth static analysis as well as dynamic analysis with the combination of HPCs in the detection of that ransomware that usually hides by implementing various obfuscation techniques (like packed or compressed programs, or indirect addressing (*Behera & Bhaskari, 2015*)). One major challenge and limitation of this research is in ransomware detection of false positives and false negatives. Consider the case of Qwerty ransomware, which uses a benign GPG executable to perform encryption. Perhaps the proposed solution would correctly detect the GPG binary when used in this way, but we suspect it would also detect it in a benign case. Since in this work we did not evaluate benign executables, it is not clear how the system performs with software that performs encryption and/or compression tasks which is the limitation of this research that will be investigated in our future work. Moreover, the collected features are related to hardware-specific environments, so if the system having the same architecture then the trained classification models are applicable as it is. However, in case the hardware environment is different (i.e., different architecture) then we have to retrain our machine learning models for that specific hardware environment. This is one of the limitations of our proposed work that the machine learning models trained on specific architecture are not portable across other machine architectures. Moreover, due to the modest dataset deep learning mechanism at present are not applicable. However, in the future, we intend to extend our dataset to implement more robust Auto Denoising Encoders, which comprises of multiple layers of neural networks and are best-known for providing good accuracy.

### Funding

The work was partially funded by Deanship of Graduate Studies and Research (DGSR), Ajman University, UAE. The funders had no role in study design, data collection and analysis, decision to publish, or preparation of the manuscript.

### Grant Disclosures

The following grant information was disclosed by the authors:
Deanship of Graduate Studies and Research (DGSR).

### Competing Interests

Muhammad Aleem is an Academic Editor for PeerJ.

## Author Contributions

- Sana Aurangzeb conceived and designed the experiments, performed the experiments, analyzed the data, performed the computation work, prepared figures and/or tables, authored or reviewed drafts of the paper, and approved the final draft.
- Rao Naveed Bin Rais conceived and designed the experiments, analyzed the data, prepared figures and/or tables, authored or reviewed drafts of the paper, and approved the final draft.
- Muhammad Aleem conceived and designed the experiments, performed the experiments, analyzed the data, performed the computation work, authored or reviewed drafts of the paper, and approved the final draft.
- Muhammad Arshad Islam conceived and designed the experiments, authored or reviewed drafts of the paper, and approved the final draft.
- Muhammad Azhar Iqbal conceived and designed the experiments, authored or reviewed drafts of the paper, and approved the final draft.

## Data Availability

   Raw data and sample python script are available as a Supplemental File.

## Supplemental Information

Supplemental information for this article can be found online at http://dx.doi.org/10.7717/peerj-cs.361#supplemental-information.

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
