# Peer review of "On the classification of Microsoft-Windows ransomware using hardware profile"

_PeerJ Computer Science, doi:10.7717/peerj-cs.361_

## Round 0.1 · original submission · Major Revisions

Please revise your paper based on the comments from two reviewers.

Reviewer 1 ·

Basic reporting

no comment

Experimental design

no comment

Validity of the findings

no comment

Additional comments

The authors proposed a ransomware detection method based on machine learning algorithms, and their model achieved accuracy of 0.97, which is good classification performance. However, some issues should be handled before publication.
1. In the end of the Abstract, you mentioned that the best accuracy was 0.97, it is suggested to present which classification algorithm achieved that accuracy.
2. How were the values of the features in Table 2 collected? Do you mean you run the ransomware in your PC and recorded the feature values, or the values were available online? Please give a more explicit description.
3. If the features are related with the hardware environment, how can the trained classification models be applied to the features from a different hardware environment?
4. The performance evaluation was based on hold-out validation, but it is preferred to use cross validation because your dataset is small, which only includes 160 cases.
5. The explanation of classification models is too short which only occupied 13 lines, and there is no novelty. You need to present the detailed description for your algorithms. Deep learning methods are popular nowadays, Please discuss the methods in 'Pathological Brain Detection based on AlexNet and Transfer Learning' and 'Detection of abnormal brain in MRI via improved AlexNet and ELM optimized by chaotic bat algorithm'. Why not use deep networks?
6. There is a miss typing in equation 4, line 362. The ‘x’ should be multiple operation.
7. Please give a comprehensive comparison of your method and state-of-the-art approaches.

Reviewer 2 ·

Basic reporting

.

Experimental design

.

Validity of the findings

.

Additional comments

Article: "On the Classification of Microsoft-Windows ransomware using hardware profile" (#52627)

The paper is quite well written; its reasoning is rather sound.

However, it seems to me that not enough technical explanations are provided:

1. It would be interesting to know exactly how the number of malware applications selected by the authors (80 non-ransomware and 80 ransomware) compares to the number of malware applications available in VirusSare database.
Are other open sources available besides VirusShare? (I do not know this, but I suspect there are...) I think the authors made a 50%-50% dataset in order to get a balanced classification problem. However, on what principles was their choice [of non-ransomware] based? Did they build an as large as possible dataset? I would kindly ask they to give us a survey of the VirusShare collection.

2. I ask myself whether the high value obtained for F-measure -- minimum 0.94, maximum 0.97 -- indicate (or not) the presence of over-fitting. (This question is in a certain measure related to the previous one: is the dataset very comprehensive or too much restricted?)

3. For how long the selected applications were ran in the "protected" mode / environment (in order to set the values for the selected HPC feature)?
And how much [varying] this time would affect the F-measures obtained for the 4 classifiers?

4. What about the parameters set for instance for Random Forests and Extreme Gradient Boosting?

How were they set?
Nothing is said in the paper about datasets used for validation (aka parameter tuning). What the authors say is that 80% of the collected set of malware from VirusShare was used for training and 20% for testing / prediction. Did they run the algorithms with implicit values of the parameters?

5. Which Extreme Gradient Boosting implementation did the authors use? (While Chen et al. 2015 publication present XGBoost, other versions of Extreme Gradient Boosting algorithms exist, some of them being more recent, and very powerful...)

4. Could we have access to the training and test datasets used/designed by the authors? (They say nothing about this issue.)


Other issues:


An overall remark:
* * *
The description of the research itself -- pages 7-11 -- is under-represented; it amounts to less than 1/2 of the whole paper.


With respect to the 'Related work' section (pages 5-7):
* * *
Most of the references there are from 2014-2017; only one is from 2020, but none from 2018-2019!
I went through the list of [titles of] papers made available by VirsuShare (the main/only resource used by the authors of the present paper!) at https://virusshare.com/research, and I saw that 52 of them contain the term 'ransomware': 10 from 2020, 15 from 2019, 17 from 2018 and 10 from 2016.
It seems to me that the authors did not update sufficiently their previous research (see L.468-469).


Some specific / "pointed" remarks:
* * *
Table 1:
It should be updated!
It contains only one reference from 2020, none from 2018-2019(!!), two from 2017 and several from 2016, 2015 and 2013. Again, it seems to me that it is too "close" to the paper published by [almost the same group of] authors back in 2017.

L.413:
The same result (0.97 F-measure) was obtained by Extreme Gradient Boosting (not only by Random Forests). Why do the authors not mention it?

L.85-86: "lack of...":
I would not agree with the authors' assertion.
(Although I am not an expert in the malware detection field, I had in the past a substantial co-operation with malware experts...)

L.232-233:
There is no contradiction here with what was written on the lines L.114-118?

L.312: Figures 2a and 2b -> Figures 3A and 3B
L.378: Figure 3 - Figure 4
L.385: Figure 4 - Figure 5

L.402: "results of 0.97" -> F-measure results of 0.97

L.413:
Do you really mean "accuracy" (twice!)
Based on the context, I would say "F-measure".

L.345-353:
I am not sure whether the notions presented here should be explained, because they are well-known in the Machine Learning community.
The same apply to lines L.374-376.
The only reason I see for including them is to make the paper self-contained.


English language issues:
* * *
L.93: "surveyed" ...what exactly?
L.99: Compare to -> Compared to
L.117: that use -> that uses
L.387: over-fitting problem -> the over-fitting problem
L.423: "to deal against ransomware to new grounds" ...What do you want to say?


Conclusion:
* * *
I would recommend a decision in the 'weak accept' to 'weak reject' interval.

---

## Round 0.2 · accepted · Accept

Thanks for your submission

Reviewer 1 ·

Basic reporting

no comment

Experimental design

no comment

Validity of the findings

no comment

Additional comments

The revised paper looks better and my comments are answered carefully. I recommend to accept it.